# New Brunswick's mental health action plan: A quantitative exploration of program efficacy in children and youth using the Canadian Community Health Survey

**Yuzhi (Stanford) Yang**[1]*, **Moira Law**[2], **Ziba Vaghri**[3]

**1** Department of Psychology, Faculty of Science, Applied Science, and Engineering, University of New Brunswick, Saint John, New Brunswick, Canada, **2** Department of Psychology, Faculty of Science, St. Mary's University, Halifax, Nova Scotia, Canada, **3** Global Child Program, Integrated Health Initiative, Faculty of Business, University of New Brunswick, Saint John, New Brunswick, Canada

* yyang26@unb.ca

## Abstract

In 2011, the New Brunswick government released the *New Brunswick Mental Health Action Plan* 2011–2018 (*Action Plan*). Following the release of the *Action Plan* in 2011, two progress reports were released in 2013 and 2015, highlighting the implementation status of the *Action Plan*. While vague in their language, these reports indicated considerable progress in implementing the *Action Plan*, as various initiatives were undertaken to raise awareness and provide additional resources to facilitate early prevention and intervention in children and youth. However, whether these initiatives have yielded measurable improvements in population-level mental health outcomes in children and youth remains unclear. The current study explored the impact of the *Action Plan* by visualizing the trend in psychosocial outcomes and service utilization of vulnerable populations in New Brunswick before and after the implementation of the *Action Plan* using multiple datasets from the Canadian Community Health Survey. Survey-weighted ordinary least square regression analyses were performed to investigate measurable improvements in available mental health outcomes. The result revealed a declining trend in the mental wellness of vulnerable youth despite them consistently reporting higher frequencies of mental health service use. This study highlights the need for a concerted effort in providing effective mental health services to New Brunswick youth and, more broadly, Canadian youth, as well as ensuring rigorous routine outcome monitoring and evaluation plans are consistently implemented for future mental health strategies at the time of their initiation.

## Introduction

Youth mental health is complex [1–4]. There is substantial scientific evidence that a plethora of physical, emotional, cognitive, and social factors interact to bolster or interfere with youths' mental health [5–7]. A variety of theoretical frameworks conceptualizing how these factors

**Data Availability Statement:** We used the 2005, 2007-2008, 2009-2010, 2011-2012, and 2015-2016 Canadian Community Health Survey, which were collected by Statistics Canada. We cannot

upload this dataset because we do not own it. However, interested parties can request any of the aforementioned datasets from Statistics Canada via https://www.statcan.gc.ca/en/microdata/pumf. Because these datasets are part of the Statistics Canada Public Use Microdata collection, Statistics Canada will share all the aforementioned datasets free of charge with any interested parties upon request. We will also provide a syntax file to interested parties for analytical transparency.

**Funding:** The author(s) received no specific funding for this work.

**Competing interests:** The authors have declared that no competing interests exist.

interact and impact youth mental health have been developed [8–10], with current perspectives now emphasizing the interdependence of a wide variety of interpersonal, familial, and societal factors affecting wellness outcomes [11, 12].

Not surprisingly, a variety of definitions for this complex construct exist [13–15]. In 2005, the World Health Organization defined mental health as a "state of well-being in which every individual realizes his or her own potential, copes with the normal stresses of life, works productively and fruitfully, and is able to make a contribution to her or his community" [16]. As the vital significance of the social determinants of health continues to be unearthed [17–20], clinicians, researchers, and policymakers have been slowly reformulating well-being; a continuum spanning from wellness to severe illness of which mental health is a constituent part [21].

Recently, the United Nations H6 Partnership Technical Working Group on Adolescent Health and Well-being, which includes representatives from each of the six major contributing organizations, including United Nations Population Fund-UNFPA, United Nations Children's Fund-UNICEF, United Nations Women-UN Women, WHO, the Joint United Nations Programme on HIV/AIDS-UNAIDS and the World Bank Group proposed a theoretical framework addressing youth well-being that incorporates five interconnected domains, namely, physical health, creating social connections, establishing safe/supportive environments, education/employment, and developing agency/resilience in youth [7, 10, 22]. In consultation with experts and referencing the scientific literature, youth experiencing mental health struggles generated an expanded definition of youth well-being; "Adolescents hav[ing] the support, confidence, and resources to thrive in contexts of secure and healthy relationships, realizing their full potential and rights." [10]. This definition included both subjective e.g., self-efficacy, and objective, e.g., poverty, constructs contained within five domains of well-being that could also be organized as preventative measures, e.g., socioemotional curriculums in schools [23] or responsive interventions, e.g., access to counselling services [24, 25].

## Adolescent Well-Being Framework (AWBF)

The domains contained within the Adolescent Well-Being Framework (AWBF) offer an array of possible but non-exhaustive targets for prevention and treatment strategies in the development and delivery of effective services for this population [10]. The five domains are physical health, social connectedness, supportive/safe environments, education/employability, and agency/resilience.

**Physical health.** Physical health is foundational to youth wellness [10]. A series of recently published meta-analyses confirm that healthy habits, including regular exercise [26, 27], adequate sleep [28, 29], and solid nutritional intake [5, 30, 31], have both short-term and lasting impacts on youth mental health and wellbeing. Food security [32, 33], health literacy [34], toxin-free environments, e.g., clean drinking water, and welcoming, accessible primary health care services were also identified as essential components of this domain [22, 35].

**Social connectedness/contribution.** Several recent meta-analyses have confirmed the necessity of the second domain in the AWBF, which includes supportive social networks, meaningful relationships, and strong family systems that foster positive attitudes and develop strong interpersonal skills, which empower youth with a strong sense of belonging [36], self-esteem [37], and self-efficacy [38]. Conversely, recognition of the growing evidence of problematic social media use and its significant negative mental health impacts must also be considered [39–45]. Minimizing loneliness is another essential target [46], as well as ensuring youth are given opportunities to significantly contribute to their community i.e., advocacy work, which develops skills in formulating thoughtful opinions, decision making and critical thinking which support health and wellness [47–50].

**Supportive/safe environment.** The third domain of the AWBF recognizes the need for safe and secure physical and emotional environments that ensure equality, fair treatment, and non-discrimination [51, 52]. Youth need to have the freedom to practice core personal, cultural, and spiritual beliefs which support physical and mental well-being [53, 54]. Policies explicitly addressing the documented negative impacts of ableism [55], racial discrimination [56] and gender/sexual orientations [57] on youth mental health and well-being must also be implemented to generate safe spaces for marginalized youth.

**Education/employability.** Education elevates the mental, physical, and social status of the person [58, 59]. Solid academic programs which provide students with skills, knowledge, and competencies that will also lead to employability are foundational for youth well-being [22]. This especially holds true for youth, particularly females, who have been placed in youth services [60]. Higher educational attainment consistently shows broad increases in cognitive abilities, interpersonal skills, and problem-solving that have direct positive impacts, such as better coping strategies and other indirect benefits, e.g., employability, for health outcomes [61] Leisure activities and time spent in recreational pursuits are also well-established predictors of mental wellness [62–64]. Ensuring youth have unfettered participation in sports teams/clubs, music lessons, and hiking/camping has documented benefits for youth well-being [65–68].

**Agency and resilience.** The fifth and final domain includes the need to develop agency and resilience in our youth [10]. Agency is when a person has confidence that their focused efforts and goal-directed behaviours will be successful [69]. Engaging learning opportunities that capitalize on students' ideas, questions, and interests [70] and civic engagement opportunities increase critical consciousness and agency among students [71]. Youth with a well-developed sense of agency will confidently face adversity as they pursue life goals [72].

Resilience is the capacity of a person to minimize or overcome the negative impacts of adversity [73–75]. Resilience can be fostered indirectly, e.g., through family systems, and directed interventions, e.g., programs. Youth who experience parental support, life satisfaction, and optimism have been found to have higher individual resilience [20, 76, 77]. Programs that focus on teaching youth coping skills, help-seeking behaviours, stress management, and mindfulness also increase resilience [78].

The AWBF framework offers all potential stakeholders i.e., policymakers, community service providers, and family members a surplus of options to organize and focus efforts to effectively intervene in youth mental health. Provided the clear meta-analytic evidence supporting the AWBF, it is anticipated that effective mental health strategies would intentionally incorporate the domains of physical health, social connectedness, supportive/safe environments, education/employability, and agency/resilience in their proposed programs' activities, targets, and outcomes.

## Mental health conditions prevalence

Despite international efforts [e.g., WHO Mental Health Global Action Programme; Department of Mental Health and Substance Abuse; 79] to improve mental health resource availability, the prevalence of mental illness remains elevated worldwide and continues to grow in several regions [80–82]. Globally, conservative estimates maintain that 12.01% of males and 12.83% of females suffer from mental disorders [83, 84]. In Canada, models estimate prevalence rates of 19.8% (or 6.8 million) of all Canadians living with a mental disorder as of 2011, with a projected increase to 20.5% (or 8.9 million) in 2041 [85–87]. The prevalence of mental disorders is notably higher amongst youth, with an estimated 23.4% (or 1 million) of Canadian children and youth aged 9–19 living with a mental disorder [85, 88, 89] These findings support a prevailing and urgent need for accessible mental health treatment resources for youth in Canada [90–92].

**COVID-19 pandemic impact.** In New Brunswick, mental health issues annually affect 30% of the youth population [93]. This statistic rose dramatically during the COVID-19 pandemic due to direct, e.g., isolation [94], and indirect pressures, e.g., parents working as front-line workers [95]. Since the COVID-19 pandemic, New Brunswick's suicide rates have risen by 40%, and hospital admissions in New Brunswick remain far above the national average [93]. Youth who were isolated due to the epidemic are now more prone to substance use issues, depression, and anxiety [72, 96, 97]. Social isolation during the COVID-19 pandemic also increased the likelihood of youth being exposed to Adverse Childhood Events (ACEs), i.e., exposure to family violence, which is known to have long-term negative mental health consequences [98]. The impact of social isolation and distancing, the public health protocols that were put in place for controlling the pandemic, was not felt the same by everyone. These negative impacts were compounded for vulnerable youth, including LGBTQ+ youth [99], Indigenous youth [100, 101] and children of households with parents working at the frontline who may have been left unattended [95].

## Delivery of services for youth

Delivery of mental health services to youth populations is challenging [102–106]. Peers naturally gravitate to informal peer support when seeking help with a variety of stressors they encounter, such as relationship challenges, conflict, depression, or suicidal ideations [8, 107–113]. If these stressors are not properly addressed, youth may engage in a variety of inappropriate coping strategies, e.g., substance abuse, that can lead to internalizing conditions, e.g., anxiety, and/or externalizing behaviours, e.g., self-harm, which can eventually escalate to suicide [114–116]. Mental health symptomology is the largest burden of disease on youth [117, 118], with high rates of relapse, negative outcomes, and societal impacts if left untreated [119–121]. With early identification and intervention, suffering can be ameliorated, quality of life can improve, and academic achievement and employability can be secured, thereby situating youth for healthy functioning adulthood [8, 102].

## Pathways to mental healthcare services for youth

Similar to the complexity of youth mental health, the pathways youth traverse to access mental health services tend to be convoluted, involving multiple failed approaches, varied contacts, wait-lists, eligibility, and undue treatment delays [9, 122, 123]. Currently, there is a relative dearth in the literature on when, how, and why youth access mental health services. In 2021, a meta-analysis conducted by Duong and colleagues [124] found the majority of mental health services for youth with elevated mental health symptoms or diagnoses were still being delivered in schools [22%] and outpatient services [21%]. More recently, peer-to-peer mentoring [125], greenspace exposure [126], one-stop-service shops [8] and digital mental health initiatives are showing promising results in terms of increasing access to mental health services [127, 128].

**Primary health care services.** Youth sometimes attempt to access mental health care through their family physician [35]; however, reluctance to involve their parents, fear of questions and/or procedures, and the lack of youth-centred approaches all reduce the efficiency of this pathway for youth mental health services [129–131]. These barriers are further compounded by the typical impediments that adults also encounter such as long wait times, low service availability, and fear of stigma [129, 132]. Hence, youth more often approach outpatient services at local hospitals for mental health services rather than their family's general practitioner [124].

**School settings.** School settings have historically delivered the bulwark of mental health services in a cost-effective and efficient manner due to their easy access to youth [133–136].

Programs vary in their content and targets [137, 138] and can be categorized as universal health promotion, e.g., socioemotional learning curriculums, or targeted interventions, e.g., programs for specific youth experiencing distress [124]. Universal health promotion programs can be classroom-based or "whole school" with a focus on behaviour policies, curriculum design, intentional support for students and staff, and highly engaged parents [139]. These programs focus on developing resilience, coping skills, problem-solving, and maintaining interpersonal relationships; they have established clear positive impacts on youths' ability to manage daily stressors, coping skills, decreased symptoms of anxiety and depression, and increased academic outcomes [140, 141]. Conversely, targeted services are delivered by school counsellors and teachers trained with listening skills and mental health first aid to provide one-on-one assistance to distressed youth [134, 138, 142, 143].

**Integrated youth health care.** Integrated services are a relatively recent innovation involving the coordination and centralization of services for physical health, mental health, and social services [8, 11, 144]. An essential ingredient to integrated youth health care is the "soft entry"; in other words, typical barriers to care are minimized and stigma nullified [145, 146]. Community services that have been proliferating in an uncoordinated manner i.e., parenting workshops, family support centers, and playgroups, can be incorporated into the one-stop-shops as part of the "soft entry" for youth and their families [147–150].

In 2019, the World Economic Forum (WEF) brought together experts to develop a Global Framework for Youth Mental Health in the delivery of services, recognizing youth mental health care needs to include a) prevention and early intervention, b) empowered youth as co-designers, c) community engagement/education, d) "soft entry" without stigma or financial barriers, e) treatment and care choice, f) family engagement and support, g) and scientific evidence as a key guide [151]. This is the future framework that needs to be implemented for optimal delivery of youth mental health services [152–155].

## New Brunswick's mental health action plan

In 2011, the Government of New Brunswick released *The Action Plan for Mental Health in New Brunswick 2011–2018* (*Action Plan*) following Judge McKee's report *Together into the Future: A Transformed Mental Health System for New Brunswick*. Representing the Government's response to Judge McKee's recommendations, the *Action Plan* consisted of seven goals: 1) transforming service delivery through collaboration, 2) realizing potential through an individualized approach, 3) responding to diversity, 4) collaborating and belonging: family, workplace and community, 5) enhancing knowledge, 6) reducing stigma by enhancing awareness, 7) improving the mental health of the population [156]. It is within this seventh goal mental health initiatives for New Brunswick youth are outlined; S1 Table provides an overview of efforts and initiatives affiliated with each goal based on the *Action Plan*'s 2015 progress report [157]. Note that the Government of New Brunswick has yet to release any further progress report on the *Action Plan* after the 2015 report.

A notable observation in the design and implementation of the *Action Plan* is the inadequacy of outcome monitoring. For instance, of the seven reports published to monitor the progress of the *Action Plan*, only three reports indicated the instruments used to monitor the progress of only three of the seven goals: funding allocation, program participation, and improvements within existing clinical programs [156, 157]. The only exception to this was the goal related to stigma reduction, which aimed for a 15% increase in the number of mentally ill individuals who would hopefully report a higher sense of belonging by 2017. Again, no corresponding monitoring instruments were proposed. Admittedly, many of the initiatives proposed in the *Action Plan* are grounded in existing evidence and practices, however, their

impact on the mental health of New Brunswicker's remains poorly understood due to a marked absence of monitoring of relevant population-level outcomes [158].

## The current study

The current study will explore the impact of the *Action Plan* on children and youth, utilizing five representative samples of New Brunswickers aged 12–19 from the Canadian Community Health Survey (CCHS) datasets. By comparing the estimated differences in mental wellness outcomes of vulnerable and non-vulnerable children and youth before and after the implementation of various initiatives within the *Action Plan*, we hope to gain a better understanding of the collective impact of those initiatives and inform future mental health policymaking. Since approximately 75% of mental illness occurs before the age of 25, mental wellness outcomes of children and youths are key metrics for evaluating the effectiveness of mental health strategies, serving as indicators of the long-term social and economic impact of poor mental health [159, 160]. Additionally, the current study also aimed to explore the utility of publicly available statistics as outcome monitoring measures of mental health strategies in the absence of Routine Outcome Monitoring (ROM) measures.

## Methodology

### Data

Data from the Public Use Micro Data Files (PUMF) of the 2005, 2007–2008, 2009–2010, 2011–2012, and 2015–2016 CCHS were used for the analyses. First administered in 2000, CCHS is a series of cross-sectional surveys conducted by Statistics Canada to monitor population health status, health determinants, and healthcare utilization. The target population of selected CCHS datasets were all non-institutionalized persons aged 12 years or older across all provinces and territories, excluding individuals living on reserves, crown lands, or certain remote regions, full-time members of the Canadian Armed Forces, and youths aged 12–17 living in foster homes.

The CCHS comprised four content components: 1) Core content consisted of questions that remained largely identical over the years and were asked of all respondents; 2) theme content consisted of questions related to specific topics asked of all respondents that changed over the years and reintroduced every two, four, or six years; 3) optional content consisted of questions selected by provinces or territories that were only asked of respondents from respective provinces or territories; and finally, 4) the rapid response content consisted of questions regarding emerging topics that were asked to all respondents, but not reintroduced in subsequent surveys [161].

In the current study, all included variables were selected from scores of questions in the core or optional content that remained consistent over the years. Prior to 2015, CCHS utilized three sampling frames for the selection of respondents: 40.5% of respondents were selected from a stratified area frame designed for the Labour Forces Survey; 58.5% of respondents were randomly selected from a stratified external administrative frame of telephone numbers in each health region; and 1% of respondents were selected from a stratified Random Digit Dialing sampling frame [161, 162]. From 2015 to 2021, CCHS utilized two sampling frames for the selection of respondents: All respondents aged 18 and over were selected from a stratified area frame designed for the Labour Forces Survey, and respondents aged 12–17 were randomly selected from a stratified Canadian Child Tax Benefit frame [161].

In order to produce representative estimates of the Canadian population, all CCHS PUMFs included person-level survey weight that corresponds to the number of persons in the population that are represented by any given respondent. The survey weight included in CCHS was

calculated based on the sampling frame. In the case where two sampling frames were utilized (e.g., the 2015–2016 CCHS), two separate person-level weights were independently calculated for each of the frames used; they were then combined into a single set of weights and calibrated to known population estimates in each health region. The resulting person-level weight accounted for non-random sampling errors that arise from both the sampling strategy and person- and household-level nonresponse [161–165]. All selected CCHS surveys before 2009 consistently achieved the target sample size of 130,000, with approximately 10,000 youths sampled [164, 165]. The 2009–2010 and 2011–2012 datasets featured a final sample size of approximately 124,000, falling short of the desired 130,000 sample size [162, 163]. And finally, the 2015–2016 dataset featured a final sample size of 109,659, again falling short of the desired 130,000 sample size [161]. The current study utilized a subset of CCHS respondents who identified as residents of New Brunswick, aged 12–19 years old. We requested all aforementioned PUMFs from Statistics Canada on September 22[nd], 2022. We were then provided copies of all requested PUMFs on September 23[rd], 2022, via Statistics Canada's Electronic File Transfer system.

## Ethics statement

We received two letters of exemption from ethics review, issued by the University of New Brunswick Research Ethics Board and Saint Mary's University Research Ethics Board, respectively, as per exemption under article 2.2 of the Tri-Council Policy Statement for research solely using publicly available statistics regulated by a government organization [166]. All PUMFs are developed from the original survey data file to meet the stringent security and confidentiality standards required by the Statistics Act prior to their release to public access. Variables most likely to lead to the identification of respondents are deleted or collapsed into broader categories. To further ensure that the security and confidentiality standards have been met, all PUMFs are formally reviewed and approved by an executive committee of Statistics Canada prior to their release [161–165]. As a result of these mechanisms, all datasets used in the current study are fully anonymized, and identification of individuals is not possible.

## Measure

**Covariates.** A range of demographic variables was controlled for in all analyses, including sex (female = reference group; male), visible minority status (no = reference group; yes), marital status (married or common-law = reference group; widowed, separated, or divorced; single), dwelling ownership (non-owner = reference group; owner), income (in $20,000 interval, from less than $20,000 to over $80,000), household size (range from 1 to 5+), and perceived physical health rated on a 5-point scale from 1 (Poor) to 5 (Excellent). Respondents were coded as visible minorities if they identified as a member of any racial or cultural group other than Caucasian [161–165]. The selection of covariates was consistent with contemporary studies on this subject [167–169].

**Vulnerable population status.** Three questions were used to determine respondents' vulnerable population status, they are: "Do you have a mood disorder such as depression, bipolar, mania or dysthymia (yes; no)," "Do you have an anxiety disorder such as phobia, obsessive-compulsive disorder or a panic disorder (yes; no)," and "In general, would you say your mental health is (excellent; very good; good; fair; poor). In the current study, the vulnerable population status was operationalized as any respondent who indicated having a formal diagnosis of anxiety or depressive disorder or rated their mental health as "fair" or "poor." The approach to the operationalization of vulnerable population status by previous psychiatric diagnoses and self-rated mental health was similar to approaches adopted by other contemporary studies on this subject [124, 170].

**Mental wellness outcome.** Five questions were used to assess the respondents' life stress, sense of belonging to the local community, life satisfaction, and utilization of mental health services. The question "Thinking about the amount of stress in your life, would you say most of your days are: not at all stressful/not very stressful/a bit stressful/quite a bit stressful?" was used to assess life stress in all selected CCHS datasets. The question "How would you describe your sense of belonging to the local community? Would you say it is: very strong/somewhat strong/somewhat weak/very weak?" was used to assess the sense of belonging to the local community in all selected CCHS datasets. The question "How satisfied are you with your life in general? Very satisfied? Satisfied? Neither satisfied nor dissatisfied? Dissatisfied? Or very dissatisfied?" was used to examine life satisfaction in 2005 and 2007–2008 CCHS datasets, starting in 2009, this question was converted to be rated on an 11-point scale from 0 to 10, with 0 being "very dissatisfied" and 10 being "very satisfied," with further addition of a derived variable converting the score back into the original five categories (i.e., 0–1 = very dissatisfied; 2–3 = dissatisfied; 4–5 = neither satisfied nor dissatisfied; 6–8 = satisfied; 9–10 = very satisfied). Finally, the questions "In the past 12 months, that is, from this date last year to yesterday, have you seen or talked to a health professional about your emotional or mental health? (yes/no) How many times?" were used to assess service utilization in all selected CCHS datasets [171–175]. The presented selection of outcome variables represented all mental wellness variables that are consistently included across all utilized CCHS datasets [171–175].

## Data analysis

All analyses were conducted using Stata17 software. Data from the 2005, 2007–2008, 2009–2010, 2011–2012, and 2015 CCHS PUMF were used for the analysis. Survey-weighted linear regression analysis was performed to investigate the research question. All selected CCHS PUMFs contain person-level weights, allowing for corrected point estimates. Screening for multicollinearity was performed using the variance inflation factor. Recall that the purpose of the current analysis is to explore the impact of the *Action Plan* by comparing the estimated differences in mental wellness outcomes of vulnerable and non-vulnerable New Brunswick youth before and after the implementation of said *Action Plan*. To achieve this, we ran a series of 2-level hierarchical linear models across five CCHS datasets to compare mental wellness outcomes between vulnerable and non-vulnerable youth within each cycle of CCHS data. Since each CCHS dataset contains a probability sample of Canadians with survey weight to adjust for non-random sampling errors, all five CCHS datasets utilized can be considered as random samples of the same population at different time periods. Thus, we can make provisional inferences regarding populational trends using calculated estimates in various CCHS datasets. The α level was set at $p < .05$. The current study has adequate statistical power to detect small effects ($d \geq 0.20$). The analytical approach is as follows:

Block 1: Mental wellness outcomes are regressed onto covariates.

Block 2: Mental wellness outcomes are regressed onto covariates and vulnerable population membership status.

## Results

### Population outcome in 2005

In the 2005 CCHS dataset, the score for the sense of belonging question was regressed onto covariates in Block 1, $F(7, 445) = 2.54$, $p = .014$, $R^2 = .053$, the vulnerable population membership status in Block 2, $F(1, 445) = 0.95$, $p = .330$, $R^2 = .055$, $\Delta R^2 = .002$. Please see S1 Table for descriptive statistics for covariates and outcomes by dataset. The vulnerable population

membership status did not predict the score for the sense of belonging question, $t = -0.98$, $p = .330$, $b = -0.16$.

Subsequently, the frequency of mental health service use was regressed onto covariates in Block 1, $F(7, 449) = 1.19$, $p = .308$, $R^2 = .056$, the vulnerable population membership status in Block 2, $F(1, 449) = 4.56$, $p = .033$, $R^2 = .128$, $\Delta R^2 = .072$. The vulnerable population membership status positively predicted the frequency of mental health service use, $t = 1.80$, $p = .033$, $b = 1.80$.

Next, the score for the satisfaction with life question was regressed onto covariates in Block 1, $F(7, 449) = 8.07$, $p < .001$, $R^2 = .138$, the vulnerable population membership status in Block 2, $F(1, 449) = 4.30$, $p = .038$, $R^2 = .150$, $\Delta R^2 = .012$. The vulnerable population membership status negatively predicted the score for the satisfaction with life question, $t = -2.07$, $p = .039$, $b = -0.29$.

Finally, the score for the life stress question was regressed onto covariates in Block 1, $F(7, 252) = 3.88$, $p < .001$, $R^2 = .104$, the vulnerable population membership status in Block 2, $F(1, 252) = 23.53$, $p < .001$, $R^2 = .159$, $\Delta R^2 = .055$. The vulnerable population membership status negatively predicted the score for the life stress question, $t = -4.85$, $p < .001$, $b = -0.77$.

## Population outcome in 2007–2008

In the 2007–2008 CCHS dataset, the score for the sense of belonging question was regressed onto covariates in Block 1, $F(7, 368) = 1.57$, $p = .142$, $R^2 = .044$, the vulnerable population membership status in Block 2, $F(1, 368) = 2.51$, $p = .114$, $R^2 = .058$, $\Delta R^2 = .013$. The vulnerable population membership status did not predict the score for the sense of belonging question, $t = -1.59$, $p = .114$, $b = -0.34$.

Subsequently, the frequency of mental health service use was regressed onto covariates in Block 1, $F(7, 372) = 1.17$, $p = .317$, $R^2 = .131$, the vulnerable population membership status in Block 2, $F(1, 372) = 11.01$, $p = .001$, $R^2 = .274$, $\Delta R^2 = .143$. The vulnerable population membership status positively predicted the frequency of mental health service use, $t = 3.32$, $p = .001$, $b = 1.83$.

Next, the score for the satisfaction with life question was regressed onto covariates in Block 1, $F(7, 372) = 4.58$, $p < .001$, $R^2 = .110$, the vulnerable population membership status in Block 2, $F(1, 372) = 1.91$, $p = .168$, $R^2 = .117$, $\Delta R^2 = .006$. The vulnerable population membership status did not predict the score for the satisfaction with life question, $t = -1.38$, $p = .168$, $b = -0.17$.

Finally, the score for the life stress question was regressed onto covariates in Block 1, $F(7, 251) = 3.52$, $p = .001$, $R^2 = .127$, the vulnerable population membership status in Block 2, $F(1, 251) = 13.08$, $p < .001$, $R^2 = .179$, $\Delta R^2 = .052$. The vulnerable population membership status negatively predicted the score for the life stress question, $t = -3.62$, $p < .001$, $b = -0.54$.

## Population outcome in 2009–2010

In the 2009–2010 CCHS dataset, the score for the sense of belonging question was regressed onto covariates in Block 1, $F(7, 334) = 2.63$, $p = .011$, $R^2 = .165$, the vulnerable population membership status in Block 2, $F(1, 334) = 0.65$, $p = .420$, $R^2 = .170$, $\Delta R^2 = .004$. The vulnerable population membership status did not predict the score for the sense of belonging question, $t = -1.45$, $p = .148$, $b = -0.18$.

Subsequently, the frequency of mental health service use was regressed onto covariates in Block 1, $F(7, 336) = 2.60$, $p < .013$, $R^2 = .054$, the vulnerable population membership status in Block 2, $F(1, 336) = 2.07$, $p = .151$, $R^2 = .061$, $\Delta R^2 = .008$. The vulnerable population membership status did not predict the frequency of mental health service use, $t = 1.44$, $p = .151$, $b = 0.54$.

Next, the score for the satisfaction with life question was regressed onto covariates in Block 1, $F(7, 336) = 5.19$, $p < .001$, $R^2 = .110$, the vulnerable population membership status in Block 2, $F(1, 336) = 2.17$, $p = .142$, $R^2 = .120$, $\Delta R^2 = .010$. The vulnerable population membership status did not predict the score for the satisfaction with life question, $t = -1.47$, $p = .142$, $b = -0.24$.

Finally, the score for the life stress question was regressed onto covariates in Block 1, $F(7, 349) = 2.21$, $p = .033$, $R^2 = .107$, the vulnerable population membership status in Block 2, $F(1, 349) = 2.69$, $p = .102$, $R^2 = .119$, $\Delta R^2 = .013$. The vulnerable population membership status did not predict the score for the life stress question, $t = -1.64$, $p = .102$, $b = -0.35$.

## Population outcome in 2011–2012

In the 2011–2012 CCHS dataset, the score for the sense of belonging question was regressed onto covariates in Block 1, $F(7, 422) = 1.97$, $p = .057$, $R^2 = .086$, the vulnerable population membership status in Block 2, $F(1, 422) = 7.41$, $p = .007$, $R^2 = .128$, $\Delta R^2 = .042$. The vulnerable population membership status negatively predicted the score for the sense of belonging question, $t = -2.72$, $p = .007$, $b = -0.51$.

Subsequently, the frequency of mental health service use was regressed onto covariates in Block 1, $F(7, 429) = 1.10$, $p = .364$, $R^2 = .087$, the vulnerable population membership status in Block 2, $F(1, 429) = 9.90$, $p = .002$, $R^2 = .182$, $\Delta R^2 = .094$. The vulnerable population membership status positively predicted the frequency of mental health service use, $t = 3.15$, $p = .002$, $b = 3.01$.

Next, the score for the satisfaction with life question was regressed onto covariates in Block 1, $F(7, 427) = 7.75$, $p < .001$, $R^2 = .158$, the vulnerable population membership status in Block 2, $F(1, 427) = 10.88$, $p = .001$, $R^2 = .188$, $\Delta R^2 = .030$. The vulnerable population membership status negatively predicted the score for the satisfaction with life question, $t = -3.30$, $p = .001$, $b = -0.32$.

Finally, the score for the life stress question was regressed onto covariates in Block 1, $F(7, 439) = 2.01$, $p = .053$, $R^2 = .111$, the vulnerable population membership status in Block 2, $F(1, 439) = 13.99$, $p < .001$, $R^2 = .171$, $\Delta R^2 = .060$. The vulnerable population membership status negatively predicted the score for the life stress question, $t = -3.74$, $p < .001$, $b = -0.70$.

## Population outcome in 2015–2016

In the 2015–2016 CCHS dataset, the score for the sense of belonging question was regressed onto covariates in Block 1, $F(7, 287) = 13.74$, $p < .001$, $R^2 = .084$, the vulnerable population membership status in Block 2, $F(1, 287) = 7.10$, $p = .008$, $R^2 = .119$, $\Delta R^2 = .037$. The vulnerable population membership status negatively predicted the score for the sense of belonging question, $t = -2.66$, $p = .008$, $b = -0.39$.

Subsequently, the frequency of mental health service use was regressed onto covariates in Block 1, $F(7, 286) = 2.45$, $p = .019$, $R^2 = .111$, the vulnerable population membership status in Block 2, $F(1, 286) = 19.31$, $p < .001$, $R^2 = .256$, $\Delta R^2 = .145$. The vulnerable population membership status positively predicted the frequency of mental health service use, $t = 4.39$, $p < .001$, $b = 1.89$.

Next, the score for the satisfaction with life question was regressed onto covariates in Block 1, $F(7, 286) = 5.52$, $p < .001$, $R^2 = .204$, the vulnerable population membership status in Block 2, $F(1, 286) = 4.75$, $p = .030$, $R^2 = .246$, $\Delta R^2 = .042$. The vulnerable population membership status negatively predicted the score for the satisfaction with life question, $t = -2.18$, $p = .030$, $b = -0.39$.

Finally, the score for the life stress question was regressed onto covariates in Block 1, $F(7, 296) = 10.90$, $p < .001$, $R^2 = .108$, the vulnerable population membership status in Block 2, $F(1,$

296) = 6.56, $p$ = .011, $R^2$ = .151, $\Delta R^2$ = .043. The vulnerable population membership status negatively predicted the score for the life stress question, $t$ = -2.56, $p$ = .011, $b$ = -0.58.

## Discussion

Youth mental health services need to be recognized as a top priority in Canada's future health care plans [90, 119]. This call for action is not new [39, 176, 177]. The current study confirms a concerted effort is still needed in providing effective mental health services to Canadian youth. The recent tragedy of 16-year-old Lexi Daken, who died by suicide six days after being sent home from a New Brunswick hospital emergency room without receiving any mental health interventions, highlights the ongoing lack of mental health services for youth. The New Brunswick Health Minister immediately called for a review of mental health crisis care [178], and the inquiry, two years after her death, noted a lack of coordination in services, a need for more education and awareness of available services, and increased community and hospital supports for mental health [179].

The *Action Plan* implemented a wide array of initiatives with the goal of improving service delivery, social integration of vulnerable populations, and training capacity of mental health professionals in the hopes of reducing stigma and improving the mental health of the population. Included in the *Action Plan* were mental wellness outcomes for New Brunswick youth [156]. However, with no program evaluation framework designed or implemented at the time of initiating the *Action Plan*, the outcomes and impacts of these efforts on the youth of New Brunswick remain unclear. The lack of evidence for improvement in the mental well-being of this vulnerable population has since been exacerbated by the healthcare system's struggle to attract, hire, and retain qualified mental health professionals [180]. Teacher shortages and dwindling resources in New Brunswick public schools [181] coupled with a diminishing number of general practitioners [182] both bulwarks of delivering mental health services to youth [86] continue to negatively impact the delivery of effective, timely services. Further, not enough post-secondary resources to train future psychologists [180] and the impact of years of pandemic related stressors [183] have also contributed to poor mental health outcomes for youth in New Brunswick.

To assess some of the impact the *Action Plan* has had on New Brunswick youth, the current study utilized cross-sectional data from five Canadian Community Health Surveys (CCHS) collected by Statistics Canada. Data was provincially collated to investigate the trend in mental wellness outcomes of vulnerable youths, i.e., those youth with a history of clinical diagnoses, by comparing their outcomes to non-vulnerable youths, i.e., those youth with no history of mental health diagnosis. Socio-demographic variables that have been identified in the published literature as social determinants of mental health, i.e., visible minority status, were included in all models of analysis as covariates [184]. Notably, the five CCHS datasets varied in the final sample size attained, with the two most recent datasets (i.e., the 2011–2012 and the 2015–2016 CCHS) falling short of the desired 130,000 sample size. Nevertheless, we expect said variations in survey sample sizes to have minimal impact on the performed analyses, as the application of the provided survey weights should account for any non-random sampling errors and minimize the impact of fluctuating sample size. The results from the regression models in this study display an overall decline in the mental wellness of vulnerable youth compared to non-vulnerable youth since the implementation of the *Action Plan*. This is of particular concern as all analyzed data was collected two years prior to any of the social disruption and adverse impacts that the COVID-19 pandemic response has had on youths' well-being [185–190].

## Mental health service utilization

This study confirmed that vulnerable youths in New Brunswick consistently accessed mental health services at a higher frequency than non-vulnerable youth during the study period. Similar findings over the past few decades have been reported when examining youth mental health service utilization [124, 191–193]. Interestingly, despite high rates of accessing services, vulnerable youth still reported higher levels of stress and lower levels of sense of belonging and life satisfaction compared to non-vulnerable youth. These are important measures as life satisfaction and sense of belonging are both well-established predictors of well-being [194–196]. Studies, like one conducted with 497 adolescents in Atlantic Canada, shed light on the apparent inconsistency between high engagement with mental health services and lower mental health outcomes. Findings suggest that quality interactions, especially those fostering a strong therapeutic alliance [197], were more indicative of future mental health improvements than merely the quantity of services received [198]. This finding of higher mental health services utilization with lower well-being outcomes warrants future investigation in New Brunswick.

A sense of belonging was the only outcome variable mentioned in the *Action Plan* that could be directly assessed using the CCHS data in the current study. On page 20 of the *Action Plan*, the stated target reads: "By 2017, increase by 15 percent the number of persons with a mental health issue who report a high sense of belonging in their communities". The data in our current study indicates this program target has not been achieved among the vulnerable youth population of New Brunswick. This is important, as youth with a lower sense of belonging are predicted to experience poorer mental health in the future [199–201]. High feelings of isolation and social loneliness [202], mistrust, and low social cohesion have also been consistently reported as contributing factors to mental decline and low community engagement among vulnerable youth [203]. Studies reporting on the adverse impacts on youth during the COVID-19 pandemic response found failing wellness indicators, i.e., loneliness, have been exacerbated since the data in this study was collected [204–206].

## Stress

Stress has a wide range of negative impacts on youth [207–209]. The current study found that vulnerable youth reported higher levels of life stress compared to non-vulnerable youth, despite reporting similar outcomes in earlier 2009–2010 CCHS datasets. Stress is not necessarily an inevitable indicator of mental illness; however, it has been associated with future psychological, physical, and behavioural problems [210]. Given the vulnerable youth in our study have a history of mental illness, e.g., depression/anxiety, or self-reported their mental health as fair or poor, increased stress experienced by this population would indicate an increase in risk for future relapse or worsening symptoms [211, 212]. Again, the COVID-19 pandemic response has since increased multiple co-existing stressors, such as physical isolation, household job loss, and learning disruptions, thereby increasing the cumulative risk for youth worldwide [17, 213, 214], including New Brunswick youth.

## Youth satisfaction with services

Client satisfaction ratings for mental health services received by youth in New Brunswick were not part of the current analysis with the CCHS data. However, the 2015 progress report [157] stated approximately 87% of clients expressed they were satisfied with the mental health services they received. The relationship between youth client satisfaction and treatment outcomes has been found to be important, with low satisfaction ratings related to increased attrition, minimal psychological improvements, and decreased likelihood of engaging future services [215–217]. Although the subjects in the 2015 progress report are not all necessarily the same

youth in the CCHS data, it presents converging evidence that many youths who received mental health services in New Brunswick during the period of our study were satisfied with services received and yet the population continues to exhibit elevated indicators of poor mental health. This high level of satisfaction with services received coupled with high levels of stress, low life satisfaction, and no reported increase in sense of belonging warrants further investigation.

Perhaps the most striking finding of this study was the lack of proximal outcome variables that could be employed to assess the *Action Plan*. The few youth-oriented objectives in the *Action Plan* included:

"1) increase the number of youth who benefit from the timely, effective, integrative approach provided by an integrated service delivery model, 2) conduct a school surveillance initiative to measure child and youth mental fitness, 3) expand the Youth Engagement Initiative throughout the province to mobilize communities and build community capacity to support youth in mental health in order to increase the number of youth with the knowledge and skills to be leaders within their respective communities, thus building individual and community resilience" [156].

However, these objectives did not have any related outcome variables in the CCHS database. Hence, the results of this study were limited in its ability to assess all anticipated impacts of the *Action Plan*. Rather only those variables in the CCHS measuring youth well-being, i.e., stress, could be used to assess goal seven of the *Action Plan*, namely, "improving the mental health of the population" [156].

Action plans and strategies are plentiful [218–223]; however, operational reviews and rigorous program evaluations are rarely conducted [224–226], and dissemination in peer-reviewed published literature is scant and riddled with ethical issues [227, 228]. Nevertheless, the few program reviews and evaluations conducted reveal much cause for concern, with one recent review found all but four Canadian provinces and territories failed to meet the minimum criteria for the WHO guidelines for child mental health policies [229]. While a broader perspective may prove insightful, the current study stopped short of conducting comparative analyses across provinces due to interprovincial variations in observed trends in vulnerable youth outcomes can be, at least partially, attributable to contextual factors specific to each province (e.g., social policy, health policy, etc.), which cannot to effectively controlled for due to limited coverage of CCHS data. Similarly, the CCHS data in this study provided insight into only one domain of the AWBF discussed in the introduction, i.e., social connectedness, by examining youths' sense of belonging and life satisfaction. Future evaluations should attempt to obtain relevant data from other sources, i.e., health records, that may provide further insight into the impact the *Action Plan* had on other predictor domains of youth well-being, i.e., physical well-being. Further, future youth mental health initiatives must consider developing, implementing, and tracking appropriate outcome measures outlined in the AWBF.

## Future directions

### Youth engagement

Calls for a coordinated effort to deliver effective mental health services to Canadian youth are not new [39, 230, 231], and without effective delivery of quality interventions, youths' well-being will continue to decline; they will experience significant individual impairment as adults, and significant societal costs can be expected [232]. First, and foremost, New Brunswick youth need to be invited and included in meaningful consultation, research, and formulation of policies and strategies related to mental health that will directly impact them [151, 233]. Youth

have a right to "participate in global [and local] conversations" [234, 235] where policy makers can identify their values, gaps in the existing system [236], and barriers to services [129, 237]. Shared decision-making (SDM) experiences of youth and mental health services are understudied [238]. Hence, youth engagement with future mental health initiatives should be subjected to ongoing evaluation to ensure youth participants' engagement results in positive outcomes that include but are not limited to satisfaction in SDM, leadership skill development, confidence, and well being [239]. Participatory action/research models necessitate the inclusion of New Brunswick youth [236, 240] simultaneously addressing one of the five AWBF domains; social connectedness/contribution that seeks to engage youth in the community in meaningful endeavours, as well as recognizing one of the pillars of the World Economic Forum's Global Framework for Youth Mental Health [10, 151].

### Routine outcome monitoring

Future initiatives need to incorporate ROM, as poor data collection, both quantity and quality, has been cited as a chronic limitation in assessing mental health services for youth [241, 242]. This lack was clearly noted in the current study. There was no commitment to a consistent collection of quality data to secure an accurate evaluation of the impact the *Action Plans's* program delivery had on youth well-being [243]. ROMs gather, cluster and analyze outcome data at local, regional or national levels [244–246] but have not been without challenges [247]. It will require thoughtful discussions and feedback from primary and secondary stakeholders to optimize future ROM efforts [248–250]. However, based on the lack of available data to assess the impact of the *Action Plan* on youth well-being a provincial strategy needs to establish and maintain local and regional ROM systems. ROMs should be planned at the client, program, and community level but must be done in consultation with primary and secondary stakeholders and specific to each initiative. For example, at the client level, ROMs collect detailed self report information from clients while they are receiving services. Information collected could include, but not be limited to, the client's perceived social functioning, current stressors, and/ or severity of symptoms. The clinician uses this information to assess their progress and may decide to change the current treatment plan based on this feedback. These databases should include variables specific to program outcomes, as well as the processes through which such outcomes are achieved, set in the strategies with personnel designated to regularly maintain, analyze, and report on the data collected. Interconnections between schools, general practitioners and emergency rooms known to be high service providers need to be established and attention to potentially disadvantaged groups such as the homeless, rural residents, and LGBTQ youth considered [251–254]. Data linkages need to be created between the healthcare system, education system and perhaps even juvenile justice system to track the individual youth in need of mental health services and assess the efficacy of the intervention at the population level. Such linkages will not only enable all parts of the system to be aware of the movements of these youth through different parts of the system, it will also help service providers to make informed decisions and prove useful in evaluating the impact of the government's efforts, such as the *Action Plan*. Such linkages must ensure adequate protection measures for privacy (e.g., such as removing personal identifiers) and model best practices in data linkage when creating such a rich database [255].

Equally important is to point out that all efforts of governments, such as policies, programs, and strategies, that are designed and implemented to resolve a public health issue must be accompanied by a rigorous monitoring and evaluation frameworks with identified timelines suitable to the intervention evaluation. Despite being designed with care and good intentions, sometimes interventions can be counterproductive to the very same situation that they were

designed to ameliorate. Without a proper evaluation, the outcome of these interventions on their target population will be concealed, and a great deal of effort, as well as public funds and even lives, may be lost in the process. Such evaluation designs must entail evaluating not just the outcomes but also the processes through which the interventions were implemented (the issues such as access, compliance with a non-discriminatory implementation of the program and provision of service for all). It is only by examining these three sets of data vis-à-vis one another, that one can see the associations between the results at the population level and the efforts of the governments.

## Special youth subpopulations

A subpopulation of vulnerable youth that will need to be considered in all future initiatives and evaluations is immigrant youth in Atlantic Canada [256, 257]. With massive, unprecedented waves of newcomers currently arriving, and forecasted to continue to arrive in Canada [30, 258] youth immigrants should also be included in planning future mental health interventions for vulnerable populations. New Brunswick plans on welcoming 7,500 newcomers a year between 2018 to 2024 [258]. Despite the healthy immigrant effect, most newcomers remain at risk for poor mental health outcomes [259, 260]. Youth even more so [261–265]. A host of established risk factors including but not limited to pre-migration experiences [256], post-migration family and school environment, discrimination, and barriers to health care, will need to be considered [266–269]. School settings as points of identification, assessment, and delivery of mental health services for these youth need to be included [270]. Governments have the same responsibilities towards these children as their Canadian-born counterparts, therefore, they must include the newcomer youth in the discussions and design of the interventions that will have an impact on their health [271].

LGBTQ+ children are another vulnerable subgroup of NB children and youth when it comes to mental health. One issue that has become a recent, visible concern, has been amendments to Policy 713 [272]. This well-intended policy, which was initially aimed to ensure the safety, privacy, and inclusion of LGBTQ+ students, received extensive amendments concerning students' ability to utilize their preferred pronouns and names in schools. In its original form, Policy 713 recognized the rights of the students to self-identify using preferred names and pronouns, constituent parts of social transitioning [273]. However, following the recent amendments, parental consent is now required for students under 16 for the use of their chosen names and pronouns in public school settings [272]. In circumstances where a student chooses not to involve their parents, they are directed to seek support from a school psychologist/social worker to assist with a plan for the eventual involvement of the parent(s). This requirement may actively undermine rights of these children under the Conventions on the Rights of the Child (CRC). Parental rights "to determine all aspects of their children's education" as outlined in the Canadian Charter of Rights and Freedoms is also being asserted [274]. These Policy 713 amendments will primarily impact the trans and gender-diverse students who haven't disclosed their gender identity at home, for fear of repercussions. Potential harms of revealing their gender identity to potentially unsupportive family members or the distress that accompanies being misgendered and deadnamed at school are research questions that need an immediate response from social scientists to provide policy makers evidence to guide relevant future decision making. The public response to the amendments has been equivocal [275–279]. The concern of the opposing groups is amplified when dismissal of children's agencies in NB instigates similar actions in other provinces of Canada [274, 280].

The CRC has a clear position on keeping a balance between child protection and child autonomy. It puts the children's evolving capacity at the forefront of the roles of children in

decision-making. The State typically defers to parents when making regular parenting choices for younger children given the child's immaturity. However, as they gain the capacity for acquiring and understanding information (Article 13 of CRC), making an informed-decision, and cohesively expressing these decisions (Article 13 of CRC), their role in such decision-making needs to be adjusted accordingly and their right to be heard play an important role in decision making in the matters that affect them (Article 12 of CRC). Such developmental processes may differ from one child to another despite their identical chronological age. The amended Policy 713 denies children their right to independently change their gender identity without parental consent, and such declaration is grounded upon the assumption that all children below the age of 16, irrespective of their individual evolving capacities, lack the skills to make such decisions on the issue so germane to their being and development.

All this while the existing evidence, presented by the 2021–2022 Student Wellness Survey [281], outlines that LGBTQ+ students of NB face increased feelings of loneliness, sleep difficulties, and lower levels of trust in communities, experiencing disparities in treatment and safety perceptions, higher levels of bullying, and feeling less supported by their families during difficult times, and more anxiety and depression. The report identifies them as one of the most vulnerable subpopulations of children for mental health issues; again, the newly amended Policy 713 may render further vulnerability and risk to their already compromised mental health.

## Explore innovative delivery

The data from the CCHS did not provide data to elucidate the nature of the delivery of mental health services to youth during the study period. Again, in future studies, other data sources from school and healthcare settings should be sought to fill this knowledge gap. Prior to the COVID-19 pandemic, youth regularly experienced barriers to care, (i.e., stigma), when seeking mental health services [282–285]. Online interventions avert these pitfalls by increasing anonymity, privacy, accessibility, self-determination, and immediacy in accessing services [286–288] Remote therapy proliferated during the pandemic response [289] and online services offer preliminary promise to address gaps in the system and smooth out inequities among vulnerable youth with access to fewer resources [290] Social networking sites-based interventions have also exhibited early evidence of being engaging, supportive and utilized by young people [291]. Similarly, the development of digital health interventions has recently proliferated [292–294] with mixed reviews [144] appearing to work better than no intervention for some disorders when used in settings that are highly supervised. However, digital health interventions have not demonstrated superior efficacy to face to face interventions regardless of the level of support offered [295]. With increases in both the prevalence of youth experiencing mental health difficulties and their engagement with online services, evaluations of these new modes of delivery should be evaluated for efficacy in reducing mental illness symptomology and distress [127].

## Limitations

The current study is not without limitations. First, the current study utilized representative samples of New Brunswick youths aged 12–19, so the observed outcomes cannot be generalized to other populations. Second, New Brunswick youths who were institutionalized, in foster care, or living on reserves at the time of data collection are excluded from the sample; thus, the result from the current study may not generalize to these populations. Third, using self-rated mental health and pre-existing diagnosis to determine vulnerable population status may exclude youths who lack insight into their mental wellness. Fourth, due to unfeasibility in combining the selected CCHS datasets, we are unable to execute a more robust analysis of temporal

trends in mental wellness outcomes. Finally, like other surveys, the CCHS is subject to self-report biases (e.g., people may under-report their use of mental health services to appear socially desirable).

## Conclusion

The results from this study suggest an overall declining trend in the mental wellness of vulnerable youths compared to non-vulnerable youths in New Brunswick, Canada, despite the implementation of an *Action Plan for Mental Health in New Brunswick 2011–2018*. This study finds vulnerable youth are consistently utilizing mental health services at a higher frequency compared to their non-vulnerable counterparts yet continue to report lower levels of life satisfaction and sense of belonging, as well as higher levels of life stress. Future initiatives to address mental health needs for New Brunswick youth need to 1) ensure services address the five AWBF domains of physical health, social connections, safe/supportive environments, education/employment, agency/resilience, 2) target high-risk youth, 3) include innovative online services and digital health interventions, 4) incorporate routine outcome monitoring at client, program and community levels, 5) establish a provincial database of relevant population-level outcome including linkages between healthcare, education, immigration, and justice systems, 6) while following principles outlined in the Global Framework for Youth Mental Health [151]. As the tide of mentally ill youth continues to rise, the allocation of scarce resources needs to target high-risk youth in accessible inter-connected settings that consistently provide quality services [144, 296–299].

## Supporting information

**S1 Fig. Statistical trend in the self-reported sense of belonging amongst vulnerable youth in New Brunswick from 2005 to 2016 (with 95% CI).**
(TIF)

**S2 Fig. Statistical trend in the self-reported mental health service utilization amongst vulnerable youth in New Brunswick from 2005 to 2016 (with 95% CI).**
(TIF)

**S3 Fig. Statistical trend in the self-reported satisfaction with life amongst vulnerable youth in New Brunswick from 2005 to 2016 (with 95% CI).**
(TIF)

**S4 Fig. Statistical trend in the self-reported life stress amongst vulnerable youth in New Brunswick from 2005 to 2016 (with 95% CI).**
(TIF)

**S1 Table. Descriptive statistics for covariates and outcomes by dataset.**
(DOCX)

**S2 Table. Overview of the New Brunswick action plan for mental health 2011–2018.**
(DOCX)

**S3 Table. Block regression result when using the 2005 CCHS.**
(DOCX)

**S4 Table. Block regression result when using the 2007–2008 CCHS.**
(DOCX)

**S5 Table. Block regression result when using the 2009–2010 CCHS.**
(DOCX)

**S6 Table. Block regression result when using the 2011–2012 CCHS.**
(DOCX)

**S7 Table. Block regression result when using the 2015–2016 CCHS.**
(DOCX)

## Acknowledgments

The authors would like to acknowledge and express a sincere and heartfelt appreciation for the support and guidance received from Dr. David Speed during the designing of this study, as well as statistical analyses of the data. The authors would also like to acknowledge the work of Ms. Rowan Hickie with the search and synthesis of summaries of some parts of the literature. We also thank Rowan for her contributions to the editing process.

## Author Contributions

**Conceptualization:** Yuzhi (Stanford) Yang, Moira Law, Ziba Vaghri.

**Formal analysis:** Yuzhi (Stanford) Yang.

**Investigation:** Yuzhi (Stanford) Yang.

**Methodology:** Yuzhi (Stanford) Yang, Moira Law, Ziba Vaghri.

**Project administration:** Yuzhi (Stanford) Yang.

**Software:** Yuzhi (Stanford) Yang.

**Supervision:** Moira Law, Ziba Vaghri.

**Visualization:** Yuzhi (Stanford) Yang.

**Writing – original draft:** Yuzhi (Stanford) Yang, Moira Law, Ziba Vaghri.

**Writing – review & editing:** Yuzhi (Stanford) Yang, Moira Law, Ziba Vaghri.

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
