## [Decision Letter · Decision Letter 0]

18 Jan 2024

PONE-D-23-36363New Brunswick’s Mental Health Strategy: A Quantitative Exploration of Program Efficacy in Children and Youth using the Canadian Community Health SurveyPLOS ONE

Dear Dr. Yang,

Thank you for submitting your manuscript to PLOS ONE. After careful consideration, we feel that it has merit but does not fully meet PLOS ONE’s publication criteria as it currently stands. Therefore, we invite you to submit a revised version of the manuscript that addresses the points raised during the review process.

We look forward to receiving your revised manuscript.

Kind regards,

Syed Hassan Ahmed

Guest Editor

PLOS ONE

Journal Requirements:

4. We notice that your supplementary [S1-S4 Figures] are included in the manuscript file. Please remove them and upload them with the file type 'Supporting Information'. Please ensure that each Supporting Information file has a legend listed in the manuscript after the references list.

Reviewers' comments:

Reviewer's Responses to Questions

**Comments to the Author**

1. Is the manuscript technically sound, and do the data support the conclusions?

Reviewer #1: Partly

Reviewer #2: Yes

2. Has the statistical analysis been performed appropriately and rigorously? 

Reviewer #1: Yes

Reviewer #2: Yes

3. Have the authors made all data underlying the findings in their manuscript fully available?

Reviewer #1: No

Reviewer #2: Yes

4. Is the manuscript presented in an intelligible fashion and written in standard English?

Reviewer #1: Yes

Reviewer #2: Yes

5. Review Comments to the Author

Reviewer #1: Reviewer feedback

Overall, the paper addresses an important topic – youth mental health and the effectiveness of mental health services in New Brunswick. Here are some critiques and suggestions for improvement:

Overall Feedback:

o Consider refining the language for greater clarity, especially for those less familiar with this specific area.

o Break down lengthy paragraphs into smaller ones for better readability.

o Limit the use of abbreviations.

o Ensure consistency in citation format throughout the paper.

o Check for grammatical and typographical errors.

Abstract:

o Ensure consistent terminology throughout, For example, In the abstract, the phrase "New Brunswick Mental Health Action Plan" is used, but later in the paper, it's referred to as the "2011 Mental Health Action Plan." Consider using consistent terminology throughout.

Introduction:

o Improve the transition between the abstract and the introduction. Consider adding a sentence or two to bridge the gap and provide a more cohesive flow between the abstract's summary and the detailed introduction.

o Briefly explain how theoretical frameworks relate to the study's objectives.

o Include more recent literature in the introduction.

o Enhance clarity and conciseness in sentence construction.

o For example: "Following the release of the action plan in 2011, two further progress reports were released in 2013 and 2015, highlighting the implementation status of the action plan" could be simplified for clarity.

o And "...impact of the 2011 New Brunswick Mental Health Action Plan on children and youth using two representative samples of New Brunswickers aged 12-19 from the Canadian Community Health Survey (CCHS) datasets." -> "...impact of the 2011 New Brunswick Mental Health Action Plan on children and youth, utilizing two representative samples of New Brunswickers aged 12-19 from the Canadian Community Health Survey (CCHS) datasets."

o Ensure consistent punctuation in phrases.

o For example: "New Brunswick Youth and youth in Canada." It should be "New Brunswick youth and youth in Canada."

o Improve sentence structure for better flow.

o For instance, in the introduction, the sentence starting with "While vague in their language..." is a bit hard to read.

Methodology:

o Enhance clarity and conciseness in various sentences.

o "Data from the Public Use Micro Data Files (PUMF) of 2005, 2007-2008, 2009-2010, 2011-2012, and 2015-2016 CCHS were used for the analyses." -> "Data from the Public Use Micro Data Files (PUMF) of the 2005, 2007-2008, 2009-2010, 2011-2012, and 2015-2016 CCHS were used for the analyses."

o "The CCHS comprised of four content components..." -> "The CCHS comprised four content components..."

o "We did not seek institutional ethics reviews for the current study as per exemption under article 2.2..." -> "We did not seek institutional ethics reviews for the current study, as per exemption under Article 2.2..."

o "In the current study, all included variables are selected from scores of questions in the core or optional content that remained consistent over the years." -> "In the current study, all included variables were selected from scores of questions in the core or optional content that remained consistent over the years."

o "The approach to the operationalization of vulnerable population status by previous psychiatric diagnoses and self-rated mental health was similar to approaches adopted by other contemporary studies on this subject (55)." -> "The approach to operationalizing vulnerable population status based on previous psychiatric diagnoses and self-rated mental health was similar to approaches adopted by other contemporary studies on this subject (55)."

o "All analyses were conducted with Stata17 software." -> "All analyses were conducted using Stata17 software."

o Provide rationale behind selecting certain variables and specifics about statistical methods.

o Include details about the regression models used.

o Discuss ethical considerations related to using survey data and ensuring participant confidentiality.

Results:

o Reduce the number of tables and figures, considering combining them.

o Ensure tables and figures do not repeat results discussed in the text.

o These tables and figures do not provide enough important and novel information to justify including that many.

Discussion:

o Include specific examples or case studies related to service delivery.

o Offer specific recommendations for improving outcome monitoring in future mental health strategies.

o Strengthen the connection between observed trends and the New Brunswick Mental Health Action Plan.

o Contextualize stressors specific to New Brunswick, such as Bill 713, struggles in recruiting mental health professionals, and the Lexi Daken Inquest.

o Discuss variations in sample sizes and their potential impact on study findings.

o Provide specific recommendations for future research directions or policy implications based on the study's findings.

Conclusion:

o Summarize key findings and their implications more explicitly.

o Clarity in language

o "Again, the COVID-19 pandemic response has since increased multiple co-existing stressors, i.e., physical isolation, household job loss, and learning disruptions, that have increased the cumulative risk for youth worldwide (57; 82; 116) including New Brunswick youth." -> "Furthermore, the COVID-19 pandemic response has increased multiple co-existing stressors, such as physical isolation, household job loss, and learning disruptions, thereby escalating the cumulative risk for youth worldwide (57; 82; 116), including New Brunswick youth."

Reviewer #2: While the article provides valuable insights into mental health patterns among young individuals in New Brunswick, there are areas where improvements could enhance its rigor and impact:

1. Clarification of Methodological Details: The article lacks clarity in explaining certain methodological aspects. For instance, a more detailed description of the rationale behind the selection of specific covariates or the reasoning behind the chosen statistical models could improve the readers' understanding.

2. Temporal Analysis and Trends: While the study examines various years of CCHS datasets, a more robust analysis of temporal trends and changes in mental health outcomes over time could strengthen the article. This would involve discussing any notable shifts or consistencies observed across different years.

3. Ethical Considerations: While the article mentions the exemption from seeking institutional ethics reviews due to using publicly available data, a brief discussion about potential ethical considerations related to using anonymized data and ensuring participant confidentiality could add depth to the methodology section.

4. Inclusion of Comparative Analyses: Incorporating comparative analyses with other regions or national averages could offer a broader perspective and help in understanding whether the observed trends are unique to New Brunswick or align with national trends. Please discuss.

5. Last but not least - the article lacks a clear presentation of the basics of the researched problem, especially the research questions with the stated hypotheses. Please complete this

6. PLOS authors have the option to publish the peer review history of their article (what does this mean?). If published, this will include your full peer review and any attached files.

Reviewer #1: No

Reviewer #2: No

---

## [Author Response · Author response to Decision Letter 0]

4 Mar 2024

February 8, 2024

From: Mr. Yuzhi(Stanford) Yang

University of New Brunswick

Faculty of Science, Applied Science, and Engineering

Department of Psychology

Email: yyang26@unb.ca

To: Syed Hassan Ahmed, Guest Editor, PLOS ONE

Dear Editors and Reviewers,

We thank you for your diligent reading and constructive feedback. As requested, please see the following for responses to the editor and reviewers. We hope this and the revised manuscript will adequately address all critiques and suggestions raised.

Editor Comments:

We have reviewed our manuscript and made the necessary changes to meet PLOS ONE style requirements.

We have now updated the Data Availability Statement.

3. Please include your full ethics statement in the ‘Methods’ section of your manuscript file. In your statement, please include the full name of the IRB or ethics committee who approved or waived your study and whether or not you obtained informed written or verbal consent. If consent was waived for your study, please include this information in your statement as well.

We received a letter of exemption from the University of New Brunswick Research Ethics Board that confirms the current study is exempted from ethics review. That information is now included in the ethics statement section (line# 313-323).

4. We notice that your supplementary [S1-S4 Figures] are included in the manuscript file. Please remove them and upload them with the file type 'Supporting Information.' Please ensure that each supporting Information file has a legend listed in the manuscript after the references list.

We have removed Figures S1-S4 from the manuscript file and double-checked the legend listed at the end of our manuscript.

Reviewers’ Comments:

Overall

We appreciate reviewers’ feedback regarding writing clarity and grammatical errors. We have made the necessary changes to the manuscript to address all writing-related issues raised. Additionally, we have reviewed and revised the manuscript to enhance the writing clarity and conciseness.

Reviewer #1

Abstract:

1. Ensure consistent terminology throughout, For example, In the abstract, the phrase "New Brunswick Mental Health Action Plan" is used, but later in the paper, it's referred to as the "2011 Mental Health Action Plan." Consider using consistent terminology throughout.

Changes are now made throughout the manuscript to ensure consistency in terminology. After its first introduction, "New Brunswick Mental Health Action Plan” is now referred to as “action plan” in the rest of the manuscript.

Introduction:

1. Briefly explain how theoretical frameworks relate to the study's objectives.

Revisions are now made at the end of the Adolescent Well-Being Framework to elaborate on the relevance of theoretical frameworks (line# 137-143).

2. Include more recent literature in the introduction.

Revisions are made throughout the introduction to reduce the amount of antiquated literature cited while citing more recent literature.

Methodology:

1. Provide rationale behind selecting certain variables and specifics about statistical methods.

Revisions were made in the Current Study, Mental Wellness Outcomes, and Data Analysis sections to provide a more comprehensive rationale for the selection of variables. Specifics regarding statistical methods were added to the Data Analysis section (line# 255-266; 362-364; 366-381).

2. Include details about the regression models used.

Details regarding regression models used are now added to the Data Analysis section (line# 373-379).

3. Discuss ethical considerations related to using survey data and ensuring participant confidentiality.

The current study utilizes publicly available statistics released by Statistics Canada. The mechanisms set forth by Statistics Canada to ensure the anonymity of survey respondents prior to the public release of data files are discussed extensively in the Ethics Statement section (line# 313-323). Upon review of our study methodology, both the University of New Brunswick Research Ethics Board and the Saint Mary’s University Research Ethics Board also concurred that no ethical issue is present and both issued a letter of exemption from ethics review.

Results:

1. Reduce the number of tables and figures, considering combining them.

By combining tables, when possible, we have now reduced the number of tables to a total of seven. However, we have retained the four figures, as combining them is not feasible.

2. Ensure tables and figures do not repeat the results discussed in the text.

After comparing the results section and the supplementary tables and figures, the only duplication between the two is the unstandardized b value for the Vulnerable Population Status in each model. After careful consideration, we felt that including b values is necessary to convey the most complete meaning of the results. Thus, the b value is retained in the revised result section.

3. These tables and figures do not provide enough important and novel information to justify including that many.

We agree; however, these tables are included as per PLOS ONE policy on reporting statistical results, which states that the full results of any regression analysis performed should be included in the submission as a supplementary file (https://journals.plos.org/plosone/s/submission-guidelines). However, as mentioned in response to comment#1 of the results section, we have made an effort to reduce the total number of tables and figures included by combining tables when feasible.

Discussion:

1. Include specific examples or case studies related to service delivery.

A new section, Explore Innovative Delivery, has been added to the Future Directions to discuss specific examples related to mental health service delivery (line# 737-752).

2. Offer specific recommendations for improving outcome monitoring in future mental health strategies.

The Routine Outcome Monitoring section of Future Directions has been revised extensively to include numerous specific and actionable recommendations for improving outcome monitoring in future mental health strategies (line# 634-676).

3. Strengthen the connection between observed trends and the New Brunswick Mental Health Action Plan.

We have expanded on our discussion of the connection between observed trends and the action plan at the end of our discussion (line# 581-613). While we have employed every available mental health outcome variable offered by the selected CCHS dataset, there remains a striking lack of proximal outcome variables that could be used to directly assess various components of the action plan, which is an acknowledged limitation of the current study.

4. Contextualize stressors specific to New Brunswick, such as Bill 713, struggles in recruiting mental health professionals, and the Lexi Daken Inquest.

A new section, Special Youth Subpopulations, is now added to Future Direction to discuss the subpopulations that are of relevance in establishing an effective mental health strategy in New Brunswick. We devoted a considerable proportion of this section to discuss matters related to LGBTQ+ youths and the implications of Policy 713 (line# 678-733). We also revised the opening paragraphs of the Discussion to discuss the Lexi Daken Inquest and the shortage of mental health professionals in New Brunswick(line# 488-497).

5. Discuss variations in sample sizes and their potential impact on study findings.

Revisions are made in the opening paragraph of the Discussion to discuss the potential impact of variation in sample sizes across used Canadian Community Health Survey (CCHS) datasets (line# 520-525). To contextualize said discussion, revisions were made in the Data section to discuss survey weight as a mechanism to address non-random sampling errors in the CCHS program (line# 294-302).

6. Provide specific recommendations for future research directions or policy implications based on the study’s findings.

The Future Directions is now considerably revised to include specific recommendations for further research, as well as policy implications arising from the findings of the current study.

Conclusion:

1. Summarize key findings and their implications more explicitly.

The Conclusion is now substantially revised to provide a more comprehensive and explicit summary of key findings and implications.

Reviewer#2 Comments:

1. Clarification of Methodological Details: The article lacks clarity in explaining certain methodological aspects. For instance, a more detailed description of the rationale behind the selection of specific covariates or the reasoning behind the chosen statistical models could improve the readers understanding.

Revisions are made throughout the Method section to provide further clarifications of methodological details.

2. Temporal Analysis and Trends: While the study examines various years of CCHS datasets, a more robust analysis of temporal trends and changes in mental health outcomes over time could strengthen the article. This would involve discussing any notable shifts or consistencies observed across different years.

The possibility of combining cycles of CCHS datasets to conduct post-estimation marginal mean comparisons was extensively explored at the early stages of the current project, as this would allow for a more robust analysis of temporal trends. A notable attempt at this is the Durham project initiated by the Durham (Ontario) health unit in 2007, which combined CCHS cycles 1.1 and 2.1 datasets to produce a report on adolescent health in the region. This was followed by a Statistics Canada report in 2009 (https://www150.statcan.gc.ca/n1/en/pub/82-003-x/2009001/article/10795-eng.pdf?st=WuYbLqRA), which provided an in-depth analysis of the feasibility and challenges of combining cycles of CCHS datasets. Ultimately, we determined that such an approach to be unfeasible for the current study as only a narrow set of CCHS can be successfully combined, and the limitations it poses outweigh the gain from a more robust comparison. Unfortunately, this also means that we can only make provisional inferences on the temporal trends.

It should be noted that this paper also serves as an academic exploration of the potential utility of using publicly available statistics as a monitoring tool for implementing and evaluating mental health strategies. The apparent limitations prompted our recommendations regarding the necessity of Routine Outcome Monitoring, which is a central theme of this paper.

3. Ethical Considerations: While the article mentions the exemption from seeking institutional ethics reviews due to using publicly available data, a brief discussion about potential ethical considerations related to using anonymized data and ensuring participant confidentiality could add depth to the methodology section.

As we have stated in the Ethics Statement, the CCHS dataset used in the current study is fully anonymized, and it is impossible to identify individual respondents. We do not foresee any ethical issues with publicly available statistics in the current study. Upon review of our study methodology, both the University of New Brunswick Research Ethics Board and the Saint Mary’s University Research Ethics Board also concurred that no ethical issue is present and both issued a letter of exemption from ethics review (line# 313-323).

4. Inclusion of Comparative Analyses: Incorporating comparative analyses with other regions or national averages could offer a broader perspective and help understand whether the observed trends are unique to New Brunswick or align with national trends. Please discuss.

Revisions are made to the end of the Discussion section to discuss the lack of comparative analysis. While comparative analyses across provinces may prove insightful, we are limited by the coverage of CCHS datasets, which has rendered interprovincial comparisons unfeasible and potentially misleading (i.e., unable to account for key contextual factors specific to provinces and territories; line# 602-607).

5. Last but not least - the article lacks a clear presentation of the basics of the researched problem, especially the research questions with the stated hypotheses. Please complete this.

Revisions are made to the Current Study (line# 257-266) and Data Analysis section (line# 370-373) to provide a clearer presentation of the researched problem.

Once again, thank you for your rigorous review of our manuscript. And we look forward to hearing from you soon.

Sincerely,

Stanford

Yuzhi(Stanford) Yang, B.A. (Hons.)

MA Student in Experimental Psychology • Department of Psychology • Hazen Hall 47

Graduate Research Assistant • Housing, Mobilization, Engagement and Resiliency Lab (HOME-RL), Faculty of Arts

100 Tucker Park Road • Saint John, New Brunswick • E2L 4L5

 506 566 0588 yyang26@unb.ca @YuzhiYang4

---

## [Editor Report · Decision Letter 1]

11 Mar 2024

New Brunswick’s Mental Health Action Plan:

 A quantitative exploration of program efficacy in children and youth using the Canadian Community Health Survey

PONE-D-23-36363R1

Dear Dr. Yang,

We’re pleased to inform you that your manuscript has been judged scientifically suitable for publication and will be formally accepted for publication once it meets all outstanding technical requirements.

Kind regards,

Syed Hassan Ahmed

Guest Editor

PLOS ONE
---

## [Editor Report · Acceptance letter]

1 Apr 2024

PONE-D-23-36363R1 

PLOS ONE

Dear Dr. Yang, 

I'm pleased to inform you that your manuscript has been deemed suitable for publication in PLOS ONE. Congratulations! Your manuscript is now being handed over to our production team.

Kind regards, 

on behalf of

Dr. Syed Hassan Ahmed 

Guest Editor

PLOS ONE